QSpec: online control and data analysis system for single-cell Raman spectroscopy

Ren Lihui
Su Xiaoquan
Wang Yun
Xu Jian
Ning Kang ningkang@qibebt.ac.cn
Bioinformatics Group of Single-Cell Center, CAS Key Laboratory of Biofuels and Shandong Key Laboratory of Energy Genetics, Qingdao Institute of Bioenergy and Bioprocess Technology, Chinese Academy of Sciences , Qingdao, Shandong , P.R. China
Wang Yong
Electronic publication date: 2014 Jun 26
Publication date: 2014
Volume: 2
Electronic Location ID: e436
Received 2014 Apr 2; Accepted 2014 May 29
Copyright: © 2014 Ren et al.
Copyright year: 2014
Copyright holder: Ren et al.
License: This is an open access article distributed under the terms of the Creative Commons Attribution License, which permits unrestricted use, distribution, reproduction and adaptation in any medium and for any purpose provided that it is properly attributed. For attribution, the original author(s), title, publication source (PeerJ) and either DOI or URL of the article must be cited.
License URL: https://creativecommons.org/licenses/by/4.0/

Keywords: Single-cell, High-throughput, Raman-activated cell sorting (RACS), Data analysis, Database, Simulation

Funding: Ministry of Science and Technology of China, Methodology Innovation Program 2011IM0301100 Ministry of Science and Technology of China, High-Tech Development (863) Program 2012AA02A707 and 2014AA021502 National Science Foundation of China 61103167 31271410 This work was supported by the Methodology Innovation Program (2011IM0301100) and the High-Tech Development (863) Program (2012AA02A707 and 2014AA021502) from Ministry of Science and Technology of China, and grants 61103167 and 31271410 from National Science Foundation of China. The funders had no role in study design, data collection and analysis, decision to publish, or preparation of the manuscript.

==============================
Single-cell phenotyping is critical to the success of biological reductionism. Raman-activated cell sorting (RACS) has shown promise in resolving the dynamics of living cells at the individual level and to uncover population heterogeneities in comparison to established approaches such as fluorescence-activated cell sorting (FACS). Given that the number of single-cells would be massive in any experiment, the power of Raman profiling technique for single-cell analysis would be fully utilized only when coupled with a high-throughput and intelligent process control and data analysis system. In this work, we established QSpec, an automatic system that supports high-throughput Raman-based single-cell phenotyping. Additionally, a single-cell Raman profile database has been established upon which data-mining could be applied to discover the heterogeneity among single-cells under different conditions. To test the effectiveness of this control and data analysis system, a sub-system was also developed to simulate the phenotypes of single-cells as well as the device features.

Introduction

All organisms on earth, including bacteria, plants and animals, derive from single-cells. Genetically identical parent cells can produce cells with different functions due to the intrinsic variation among the individual offspring cells in gene expression and gene regulation. Microbiologists are especially interested in single-cell techniques because most microorganisms (>99%) have not yet been cultured in the lab (Amann, Ludwig & Schleifer, 1995; Rappe & Giovannoni, 2003). These uncultivated microorganisms contain a large amount of functional genes and play crucial roles in natural ecosystems through various ways such as global warming (Monson et al., 2006), food security (through maintaining soil health and promoting plant growth) (Xia et al., 2011), and environmental bioremediation (Beja et al., 2000).

The monitoring of microbial single-cells in vivo during the time course is an effective method to analyze the adaptation of a population to changing conditions, such as nutrient supply or stress exposure. Notwithstanding culminating evidences for varies adaptation diversities among individual “population or community” members, such endeavors have only been undertaken recently due to enormous technical challenges. Regardless of these obstacles, such studies hold great promise to provide substantial new insight into fundamental physiological processes in microorganisms as well as to accelerate the development of superior strains for industrial biotechnology.

Single-cell technologies, such as FACS analysis and the more recently developed RACS (Li et al., 2012), are capable of detecting phenotypic heterogeneities in cellular population. Raman spectroscopy is an especially powerful analytical technique which has already been used in the study of single-cells. Raman spectroscopy is based on inelastic scattering of photons following their interaction with vibrating molecules of the sample. During this interaction, photons transfer (Stokes)/receive (Anti-Stokes) energy to/from molecules as vibrational energy. Thus, the energy change of the scattered photons corresponds to the vibrational energy levels of the sample molecules. For more detailed description of the physics of the Raman spectroscopy please refer to Ferraro (2003). Raman micro-spectroscopy can provide useful biochemical information regarding live cells, therefore has a wide application area including environment monitoring, healthcare, bioenergy, etc.

Recently, single-cell based Raman spectroscopy profiling (a light scatter analysis technique) has become highly appropriate at resolving the dynamics of cells at individual level by recording and comparing single-cell Raman spectra, yet the discrimination power of the Raman profiles is not particularly strong at distinguishing marginally different phenotypes. Nevertheless, RACS has several advantages over the classical fluorescence-based sorting (Li et al., 2012). It can survey natural microbial communities or study gene expression variance in cells of the same genotype without artificial interference such as external tagging of cells or fluorescent protein insertion (Wagner, 2009).

The RACS system automates the delivery, manipulation, analysis and sorting of single-cells from a continuous flow of cell samples. It enables the separation of cells according to their intrinsic chemical ‘fingerprint’ with minimal pre-treatment, thus cells are potentially viable after sorting (Huang, Ward & Whiteley, 2009). The isolated cells can then be further processed on a chip for cultivation or DNA amplification (Huang, Ward & Whiteley, 2009). Tweezers or microfluidic chips-based techniques combined with Raman micro spectroscopy could be used for tumor identification (Huang et al., 2004; Wlodkowic & Cooper, 2010b), cancer recognition (Wlodkowic & Cooper, 2010a) and stem cell research (Pascut et al., 2011; Wang et al., 2005), etc. Given that the number of single-cells to be analyzed and isolated would be massive in most experiments, the power of Raman profiling techniques for single-cell analysis would be fully utilized only with the accompaniment of high-throughput and intelligent online control and data analysis system.

In this work, we describe our approach for RACS system intelligent control and high-throughput data analysis in the following order: (1) Establishment of an automatic high-throughput process control system QSpec (http://www.computationalbioenergy.org/qspec.html) that could support the full cycle of single-cell phenotyping: instrument control (including RACS platform control and microfluidic device control), single-cell image analysis, single-cell Raman profiling, single-cell profile comparison, etc. (2) Based on this system, a single-cell Raman profile database was established based on which some database search and data-mining works were performed to discover the heterogeneity among cells under different conditions and at different time-points during differentiation. (3) To test the effectiveness of the whole control and data analysis system, we had also created a simulation system which can simulate single-cell features as well as device features, and tested the QSpec system with it. (4) The whole QSpec system is put to test on the prototype of real single-cell Raman spectrum analysis platform. QSpec is an easy-to-use, fully-customizable, memory-efficient and fast software package that could be run on a desktop computer.

Materials and Methods

The QSpec software system is designed to be coupled with the RACS system for single-cell manipulation. Therefore, the RACS hardware is the foundation based on which QSpec software is designed. Figure 1 shows the RACS hardware, which consists of a microscope, a Raman excitation laser, optical grating, a spectrograph, a microfluidic device and some accessory modules. The microscope objective has a numerical aperture (NA) of approximately 0.9 to produce a sharp focus to trap micro-particles suspended in solutions. The arrangement shown in Fig. 1 is in a signal acquisition format, where the Raman scattering is collected through the same lens as the excitation. Two lasers with different output beam of wavelength are used as the trapping laser. The choice of wavelength is dictated by the following considerations: The efficiency of Raman scattering exhibits the λ−4 wavelength dependence, indicating the desirability of using short wavelengths. However, short wavelengths result in increased propensity for laser-induced photo damage. The use of longer wavelengths also offers a potential advantage of reducing fluorescence effects that compete with the weak Raman signals. However, longer wavelengths also lead to inescapable problems that are related to the efficiency of CCD-based photon detectors that are readily available at present. The best available detection efficiency of ∼50% is obtained over the wavelength range 500–800 nm (Snook et al., 2009). The corresponding value around 1 µm wavelength is 10% while at 1064 nm it is almost zero. Therefore, in our RACS system, our choice of 532 nm and 785 nm as the Raman excitation wavelength appears to be the better utilitarian compromise. Manipulation of the living cells is achieved simply by means of microfluidic device and controllable x–y–z platform.

Figure 1 Schematic representation of RACS hardware set-up.

All hardware are annotated at sides, and arrows indicate laser pathways.

Base on the RACS hardware, we have developed an automatic high-throughput process control and data analysis system QSpec (Fig. 2). QSpec was implemented using C++. To automate and streamline the fast data process, QSpec takes advantage of multi-thread computation. QSpec system consists of 5 major components: (A) instrument control (including Raman profiling platform control and microfluidic device control), (B) single-cell image analysis, (C) single-cell Raman profiling, (D) single-cell database update and (E) database search. Figure 3 is a screenshot of the QSpec system in which all of these five components and related results were shown simultaneously on the same screen.

Figure 2 Framework of QSpec.

(A) Instrument Control, (B) Image Analysis, (C) Raman Profiling, (D) Database Update and (E) Database Search.

Figure 3 Screenshot of user interface for QSpec.

The single-cell entry in the list, its coordinate, its image and the Raman spectrum were shown on the same screen.

Automatic control and single-cell phenotype extraction

(A) Instrument control: For the instrument control, we designed two major components: Raman profiling platform control, and microfluidic device control. Figure 4 shows the operation interface of the RACS platform control and microfluidic device control. RACS platform control is for adjustment of the parameters of the spectrometer, the motor, the laser device, etc. By means of these controls, we achieved the fully adjustable single-cell Raman spectrum collection and signal processing. The function of microfluidic device control is for the parameter adjustment of electromagnetic valve to facilitate the sorting of the cells which we are interested in.

Figure 4 The instrument control parameter setting interface for (A) RACS platform control and (B) microfluidic device control.

In (B), the cell sorting is started based on single-cell Raman spectra, and the electromagnetic valve could be turned on and off for cell sorting if (i) the ratio of one peak intensity over another is greater than a threshold as defined or (ii) the difference of one peak intensity minus another is greater than a threshold as defined (as annotated in blue rectangles).

(B) Image analysis: For image analysis, single-cell images are extracted through a series of steps. Firstly, single-cell was identified from an image (from microscope) containing many single-cells, which could be considered as a classical pattern recognition problem, and then Sobel/Prewitt algorithm (Sobel, 1978) with default parameters was used to recognize each of the single-cells’ boundary from the relatively large image. After this, a conservative approach have been adopted, in which single-cells with completely clear boundaries (defined by Sobel/Prewitt algorithm) and do not overlap with other cells were considered as positive single-cell images. Other single-cells could either be single-cells with abnormal shape, or single-cells with overlapping images, or noises, based on which the Raman spectrum might not be retrieved correctly. In this way, we could achieve high specificity while relatively lower sensitivity in single-cell image extraction.

(C) Raman profiling: Raw Raman spectrum extraction is a routine with fixed parameters such as laser focus position, focusing time, etc. (Lewis & Edwards, 2001). After the Raman spectrum for a single-cell was obtained, we designed a routine for quality control and filtration process through a series of steps: Baseline correction, smoothing, Fourier Transform for correction and normalization of the Raman spectrum (as below):

Baseline correction allows background in a spectrum to be subtracted, to yield a spectrum with zero baseline. The correction can be applied to a single spectrum or a multidimensional spectral array.

The smoothing function allows spectra to be smoothed and converted to first and second derivative functions. Typically these functions allow spectral quality to be improved after acquisition.

The Fourier Transform function allows smoothing of a spectrum based on direct Fourier data transformation, applying the filteration and apodization functions. The spectrum is converted into its real and imaginary Fourier functions, which essentially represent the spectrum as a combination of wave patterns of varying frequency. Smoothing can be applied by removing high frequency contribution (corresponding to noises) and leaving medium and low frequency contribution (corresponding to Raman peaks).

Database and data comparison

After single-cell’s Raman profile extraction and adjustment, the Raman spectrum will then be searched against the Raman spectra database for fast classification and sorting. These would involve (D) database update and (E) database search (Fig. 7).

The database is organized as a two-tiered structure: the raw database contains all single-cell phenotypes (Raman spectra and images) collected from the RACS system, and the refined database was created based on the raw database, containing only representative single-cell phenotypes that were of high quality as follows: (1) Selection based on significance: If the intensity of a particular peak is higher than the defined peak intensity, or if a specific peak has appeared, then it would be automatically stored in the refined database; (2) Selection based on manual inspection: Raman spectra could also be manually selected and inserted into the refined database, based on random selection or targeted selection procedure. For example, in the case of Saccharomyces cerevisiae BY4743 in Table 2, when the peak at about 1003 cm-1 appears and the signal noise ratio is over 3, it would be deposited into the refined database.

A package of data analysis tools, which includes support vector machine (SVM) (Vapnik, 2000) and Euclidean Distance (ED) (Danielsson, 1980), has also been designed for the comparison and interpretation of the data in the database, as well as for cell classification (Amantonico et al., 2010; Whitaker & Walt, 2007) and effective database search (against the refined database). Additionally, a programmable portal has been created to link these single-cell phenotypes to their related omics data, so that phenotype–genotype association studies could be conducted.

For SVM analysis, the kernel function plays a key role in solving classification problems because many such applications are not linearly separable in their original dimensional space. By applying a kernel transform K, the input data vectors are mapped into a higher-dimensional space. In this space, the mapped data vectors could be linearly separable or have improved separability. The Radial Basis Function (RBF) kernel is commonly considered as the most powerful, so it was applied in this work. RBF kernel is defined as (1) Kxi,xj=exp−γ‖xi−xj‖2for γ>0

where xi, xj are two training objects in the dataset.

Using this kernel, the radial width γ has to be estimated. The optimal value of γ is found after a grid search which only needs to be performed once for a given classification task.

The simulation system: simulation of cell features and device features

Currently, there is still a lack of single-cell phenotype information from different aspects: firstly, the types of single-cells examined is limited; secondly the number of single-cells collected is far from enough; thirdly the single-cell Raman spectrometry is quite a new technology so that related phenotypes are largely lacking. In order to compliment the current little information about real single-cell phenotypes, the simulation system was established. This systems could simulate the features of single-cells as well as the device features in the following aspects: (a) multiple single-cell phenotypes, including realistic single-cell images, Raman spectra, positions and the effects of in vivo single-cell dynamics (such as the Brownian motion) are simulated, and (b) virtual platform operation interfaces, focus adjustment and Raman spectra capture functions (http://www.computationalbioenergy.org/qspec-simu.html). Figure 5 illustrates the framework of the simulation system, which consists of several major components: simulation of phenotype (images, Raman profiles, positions) and simulation of system control.

Figure 5 The Framework of simulation system and its connection with QSpec system.

In single-cell simulation part (top), the supervised method (based on training dataset) is used for the simulation of single-cell’s image, Raman spectrum and position. In the QSpec system (bottom), the simulation of platform shift has also been implemented.

For the simulation of single-cells features, three main simulation processes are deployed based on the input cell type, cell density, number of cells, etc. They include:

(1) Given cell density and the number of cells to be simulated, the cell positions are all randomly generated and positioned onto the plate.

(2) Given a specific cell type, a large number of single-cell images were used for training (by k-Nearest Neighbor (KNN) method) a simulation model for different types of strains under different conditions, and this simulation model is used, together with a random noise simulation process, to produce the simulated single-cell images for specified types of single-cells.

(3) Given a specific cell type, the simulated Raman spectra are generated in a similar way as for simulated single-cell images. A large number of Raman spectra were used for training (by KNN method) a simulation model for different types of strains under different conditions, and this simulation model is used to produce the simulated single-cell Raman spectra for specified types of single-cells.

For the device features, the movements of laser focuses are also simulated in the same pattern as for real Raman spectroscopy. Based on the resulting single-cell Raman spectroscopy and images from this simulation as input, the QSpec would run exactly the same as on real single-cells.

The simulation system can simulate a single-cell’s dynamics and obtain spectral information. Therefore, it would facilitate easy configuration and diagnosis of various modules in QSpec system.

Results

Materials and assessment methods

The RACS system that we used is a prototype that consists of Raman spectroscopy, single-cell sorting, microfluidic device module and QSpec control and data analysis module.

QSpec runs on Windows operating systems (Windows XP and later versions). We provide user instructions and sample input files for ease-of-use of the entire package (http://www.computationalbioenergy.org/qspec.html). We also provide a set of scripts to test the accuracy and speed of QSpec.

To assess the effectiveness of QSpec on high-throughput single-cell phenotyping, we applied QSpec on thousands of real algae, yeast and bacterial single-cells under different conditions and at different time-points based on RACS-1 prototype system (details in Table 2). For algae single-cells, we selected Nannochloropsis oceanica IMET1, and analyzed their different single-cells though a time-course under N-depletion/repletion condition. For yeast single-cells, we have selected Saccharomyces cerevisiae BY4743, and analyzed the single-cells at stationary phase. The bacterial single-cells that we have used are from Streptococcus sanguinis at stationary phase.

The simulated data are generated based on KNN-based training model from a large number of Raman spectra for different types of strains under different conditions (details in “Methods”).

The effectiveness of QSpec is measured by sensitivity. Sensitivity is defined as the percentage of correct cells that QSpec has recognized: (2) Sensitivity = # Correctly recognized cells# Correct cells.

Accuracy test on the instrument control module

The manual operation for single-cell profiling is not only time-consuming but also makes it difficult to choose the single-cells with specific properties in a high-throughput manner. In QSpec, the control system is fully automatic, so it could facilitate high-throughput single-cell analysis. This characteristics is extremely useful for massive single-cell extraction and analysis. As QSpec could locate single-cells by using image detection technology with low error-rate, it is more effective to choose the cell automatically.

To test the accuracy of this fully automatic process, we used 10 single-cells of Saccharomyces cerevisiae BY4743 under normal conditions to test the reliability of automatic measurements. The experiment parameters were: microscope objective number 50, laser wavelength 532 nm, exposure time 5 s. Results have shown that through instrument control, when performing single-cell image analysis and single-cell Raman profile analysis, the differences between automatic and manual measurements are very small (Fig. 6 shows the results for two single-cells), indicating that the automatic control and analysis system is indeed reliable and feasible.

Figure 6 The comparison of Raman spectra obtained by un-normalized automatic and manual measurements.

(A) and (B) showed the Pearson correlations for all peaks measured by automatic (X-axis) and manual (Y-axis) methods for two single-cells. (C) and (D) showed the real Raman profiles measured by automatic (blue) and manual (green) methods for the two single-cells in (A) and (B), respectively.

Accuracy test on a simulation system

The simulation system first created simulation cells that contain Raman profile and image information. The QSpec system would extract these single-cell phenotype information in the same way as on real cells. This simulation system would be used to evaluate the sensitivity of QSpec in extracting the images and Raman spectra of single-cells.

We tested the effectiveness of the simulation system and analyzed simulation cells based on simulated Saccharomyces cerevisiae BY4743 single-cells with different configurations, focusing on the simulated single-cell position, image and Raman spectrum (refer to “Methods”, Fig. 5). Firstly, the positions of simulated single-cells were extracted based on single-cell image analysis. From this step, we could obtain the sensitivity of image analysis (Table 1). Then the Raman spectrum for each cell was obtained from previously identified single-cell positions. However, due to the overlap of multiple single-cells, Raman signal might not be obtained from previously positioned single-cells. Results on sensitivity of Raman profiling (Table 1) have shown that the sensitivity for Raman profiling was slightly lower than that of image analysis.

Table 1 Sensitivity of single-cell image and Raman spectrum extraction based on simulated dataset.

Simulated cell	# cells
(window size fixed)	# correctly
recognized cells	Sensitivity of
image analysis	Sensitivity of
Raman profiling	
Saccharomyces cerevisiae	100*5	475	95.0%	91.0%	
Saccharomyces cerevisiae	100*10	956	95.6%	95.0%	
Saccharomyces cerevisiae	200*5	932	93.2%	91.3%	
Saccharomyces cerevisiae	200*10	1931	96.5%	96.1%	

Table 2 Accuracy for single-cell phenotype extraction based on yeast and bacterial cells under various test conditions.

Cell	Test
condition	#
cells	# correctly
recognized
cells after
image analysis	Sensitivity of
image analysis	# correctly
recognized
cells after
Raman profiling	Sensitivity of
Raman profiling	
Saccharomyces cerevisiae BY4743	Tube	115	82	71.3%	48	41.7%	
Saccharomyces cerevisiae BY4743	Slide	96	75	78.1%	62	64.6%	
Streptococcus sanguinis	Tube	80	43	53.7%	16	20.0%	
Streptococcus sanguinis	Slide	73	41	56.1%	33	45.2%	

Results based on the RACS prototype instrument

In order to verify the performance of QSpec system on RACS-1 prototype, more than two hundred samples of Saccharomyces cerevisiae BY4743 and Streptococcus sanguinis single-cells were tested (Table 2). Through the automatic control we measured different growth conditions of cells and got more sets of Raman spectra. Figure 7 is a screenshot of QSpec analysis for cerevisiae, in which five out of six single-cells’ images and Raman spectra have been correctly extracted.

Figure 7 The screenshot of QSpec analysis for Saccharomyces cerevisiae.

The background shows the single-cell images and their positions on the plate under microscope, and at foreground the Raman spectra for 5 single-cells are shown.

Then we tested the effectiveness of QSpec single-cell extraction and analyzed single-cells from cerevisiae BY4743 and sanguinis populations on large scales. Due to cell irregularity, Brownian motion, and overlap of multiple single-cells, positional deviation is found when pointing lasers onto some of the single-cells after identifying them (there is a time-lag between these two processes) using QSpec. Under such situations, Raman signal cannot be obtained for previously positioned single-cells. Thus in theory the results on real single-cells would be worse than those on current simulated single-cells (Table 1). And in practice this was actually the case (Table 2).

Results based on different strains of single-cells under different test conditions (tubes and slides) (Table 2) have also shown that relatively high sensitivity of image analysis could be achieved, while the sensitivity for Raman profiling was slightly lower. Additionally, results based on “single-cells in tube” are generally worse than those based on “single-cells on plate”, probably due to Brownian motion and other factors. Moreover, the sensitivity of both image analysis and Raman profiling was lower than that based on simulated single-cells, indicating that the above-mentioned Brownian motion and other factors might affect the accuracy of single-cell phenotype extraction.

Single-cell Raman profile database and data comparison

Another important part for the whole QSpec platform is a single-cell Raman profile database, which would facilitate the comparison of different profiles as well as data-mining for bio-markers. Several basic features were considered in our initial attempt in building a prototype of this database: (A) Project information: Cell ID, Project ID and Date; (B) Sample preparation: Name, Temperature, Shaking, OD and so on; (C) Instrument parameters: Laser, Filter, Objective, Grating, etc.; (D) Cell information: Image, Raman spectrum, Coordinates, etc. (Fig. 8). Combined, these features could potentially answer fundamental questions such as the similarities and differences among single-cells with complex spatial–temporal relationship, as well as the co-localization of the single-cells.

Figure 8 The structure of the prototype of the single-cell Raman profile database.

It is composed of 4 major parts: (A) cell identity information, (B) culturing experiment information for cells, (C) Raman spectroscopy experiment information for cells, (D) cell image, Raman spectrum and position information.

Based on the automatic single-cell extraction and the massive number of single-cells, a database including single-cell Raman profiles as well as images and position information was created, so that single-cell Raman signal comparison and other data-mining could be performed. The current raw database contains more than 100,000 single-cell Raman spectra, and the refined database has 12,011 single-cell Raman spectra. These single-cells come from 14 strains including: Nannochloropsis oceanica IMET1, E.coli DH5α, Schizochytrium SR21, Saccharomyces cerevisiae BY4743, Actinomyces viscosus C505, Enterococcus faecalis TCC29212, Porphyromonas gingivalis W83, Streptococcus mutans UA159, Streptococcus sanguinis ATCC49425, Staphylococcus aureus ATCC6538, Staphylococcus epidermidis ATCC 12228, Chlorella pyrenoidosa, Thermoanaerobacter sp. X514, Chlamydomonas reinhardtii. These microbial strains represent a wide range of species from different growth conditions, and thus could serve for our analysis of single-cell phenotypes.

For each pair of randomly selected species, we have randomly selected 100 single cells from the pool of all single-cells for the two species, and performed PCA analysis for each run. The same analysis was repeated for 100 times. Based on the average performance, PCA analysis could achieve the “separation error rate” (defined as the number of wrongly assigned single-cells, divided by the number of total cells) of 5.8%. Figure 9 showed the separation result of Staphylococcus aureus ATCC 6538 and Staphylococcus epidermidis ATCC 12228 cells by one run of PCA analysis, representing 50 randomly selected cells for ATCC 12228 and 50 randomly selected cells for ATCC 6538, two Staphylococcus strains that are phylogenetically very close. Based on their Raman spectra, the separation error rate of 5% could be achieved. These results indicated that the different cells in this single-cell database could be easily distinguished from the other cells based on their Raman spectra. Moreover, based on these results on randomly selected sets of single-cells from different strains, we concluded that the quality of the refined database is relatively high.

Figure 9 PCA analysis results based on single-cell Raman spectra for Staphylococcus aureus ATCC 6538 and Staphylococcus epidermidis ATCC 12228 in a refined database.

Single-cells from different strains can be clearly separated by PCA analysis based on their Raman spectra.

Single-cell Raman profile comparison

We also tried to search single-cell Raman spectra against the refined Raman spectra database using SVM and Euclidean Distance methods. The analysis of Raman spectra was performed in two steps: (1) preprocessing of the spectra and (2) search by using SVM or Euclidean Distance to compare the query Raman spectrum against those in the database.

For Raman spectrum database comparison and search, 120 randomly selected single-cells from Nannochloropsis oceanica IMET1 were used as queries against the current refined database with 12,011 single-cell Raman spectra, and repeated the query selection and search process for 10 times. The accuracy is defined as the number of single-cells that matched to the correct strain, divided by the number of all query single-cells, and we could achieve the accuracy of 93.3% based on SVM, and 82.9% based on Euclidean Distance in this study (Fig. 10). Additionally, to measure the effect of increasing database size on accuracy and efficiency in database search, we randomly selected 6 subsets of refined database with increasing sizes (with sizes of 810, 2,100, 3,500, 5,600, 6,700 and 8,451), and performed searches against these subsets. For SVM analysis, we have used nu-SVC for the type of SVM and RBF for Kernel type. Results have shown that SVM is better than Euclidean Distance on accuracy, but it required more time (Fig. 10). This might be due to the fact that the database size is not very large. With the increasing database size, the time taken for model-based SVM is expected to be shorter than that based on Euclidean Distance.

Figure 10 The comparison of (A) search accuracy and (B) search time based on SVM and Euclidean Distance methods.

Each data point represents the average value based on 20 single-cells as queries.

Discussion

In this work, we have developed a control and data analysis system QSpec for high-throughput Raman activated cell sorting (RACS) platform. QSpec could greatly facilitate high-throughput single-cell analysis. It is designed not only for automatic and intelligent control of the RACS platform, but also for data analysis of the extracted single-cell Raman spectra and images. Results based on simulated single-cells and real algae and yeast single-cells proved that QSpec can be used for accurate single-cell phenotype extraction, as well as for single-cell phenotype data analysis. With the rapid development of single-cell analytical equipment as well as the needs for single-cell omics research (Zong et al., 2012), the QSpec control and data analysis system would facilitate fast phenotype screening and sorting, and thus would be indispensable.

Additionally, the Raman spectrum database and its companion data-mining system have enabled not only the storage, but more importantly online single-cell profile comparison, thus making high-throughput single-cell phenotyping and screening possible.

Moreover, for the simulation system, though extensive improvements are required to mimic real cells and their dynamics, it would be essential for the development of more sophisticated and advanced single-cell manipulation systems (microfluidic devices, signal retrieval systems, machine learning methods to simulate the cells, etc.). Thus its further update and optimization, which could make simulated cells to behave more likely as real cells, by either supervised or unsupervised learning methods, would greatly facilitate the development of future version of QSpec.

Additional Information and Declarations

Competing Interests

Author Contributions

The authors declare there are no competing interests.

Lihui Ren conceived and designed the experiments, analyzed the data, contributed reagents/materials/analysis tools, wrote the paper, prepared figures and/or tables, reviewed drafts of the paper.

Xiaoquan Su performed the experiments, contributed reagents/materials/analysis tools, reviewed drafts of the paper.

Yun Wang performed the experiments.

Jian Xu wrote the paper, reviewed drafts of the paper.

Kang Ning conceived and designed the experiments, contributed reagents/materials/analysis tools, wrote the paper, prepared figures and/or tables, reviewed drafts of the paper.

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
