# Peer review of "QSpec: online control and data analysis system for single-cell Raman spectroscopy"

_PeerJ, doi:10.7717/peerj.436_

## Round 0.1 · original submission · Minor Revisions

· Academic Editor

Minor Revisions

Two review reports are at hand. Both reviewers agreed that the authors addressed an important problem and request further revision before acceptance.

·

Basic reporting

No Comments

Experimental design

No Comments

Validity of the findings

No Comments

Additional comments

The authors present QSpec, an automatic system to perform high-throughput single cell phenotyping, to establish a single-cell Raman profile database, to simulate single-cell features from the profile database, and to perform single cell image analysis and database searching and comparison using several machine learning methods like SVM and KNN. The simulation study and analysis on algae, yeast, and bacterial single-cells under different conditions show relatively good performance of the system.

Single cell phenotyping is very important for early cancer diagnosis, understanding of cell differentiation and development, and revealing the microbial world. And thus an automatic system like QSpec would largely facilitate researches on these areas and present a starting point for more sophisticated and advanced single-cell manipulation systems.

Minor concerns:
<1> In page 10, (B) Image analysis, line 6, “those with suitable sizes are considered positive single-cell images”. I think it is better to describe the criteria for “suitable size”.

<2> In page 10, (B) Image analysis, line 10, “only for single-cells with clear images”. Again what is the criteria for “clear images” and what is the portion of “clear images” as I expect a lot of cell overlapping due to the large cell numbers? The study could be improved if the author could propose some machine learning methods to deal with overlapping images in the discussion parts as a future direction.

<3> In page 10, (C) Raman profiling, the authors mentioned quality control and normalization of raw data, but the details are missing. If it is a routine, I think the authors should cite some references.

<4> The authors implemented SVM and KNN for the comparison and interpretation of data. However, in the real data analysis, they compare SVM with Euclidean Distance. There is some inconsistency between Method and Result parts.

<5> The performances of the method on simulation data and real data are quite different as revealed by Table 1 and 2. I can understand that it is very difficult to simulate the real cells, however, it would be beneficial if the author could suggest some future directions to improve the simulation data. E.g. how to simulate the Brownian motion and other factors like cell overlapping.

<6> In page 24, it appeared to me that the authors only performed the random selection of 100 single-cells for a single time to calculate the separation error by PCA. To avoid randomness, a better solution is to perform such analysis for say 100 times and average the error rate. The same problem appears in page 25, e.g. “120 randomly selected single-cells”.

<7> They are a few typos.
• Page 3, line 6, “(Amann et al. 1995; Rappe & Giovannoni 2003) .” delete the gap before “.”.
• Page 5, line 8, “with the accompanying high-throughput and ”, should be “with the accompany of high-throughput”
• Page 6, line 2, “memory-efficient, fast software package”, should be “memory-efficient, and fast software package”

Reviewer 2 ·

Basic reporting

No comments.

Experimental design

In general, the experimental design and description is appropriate. The authors should include some detail about the source and culture of the bacteria used.

Validity of the findings

No comments.

---

## Round 0.2 · accepted · Accept

· Academic Editor

Accept

The manuscript has been greatly improved. I suggest its publication.